# Optimization of Exocrine Pancreatic Insufficiency in Pancreatic Adenocarcinoma Patients

**DOI:** 10.3390/nu16203499

**Published:** 2024-10-15

**Authors:** Jaclyn V. Moore, Charles R. Scoggins, Prejesh Philips, Michael E. Egger, Robert C. G. Martin

**Affiliations:** 1Norton Cancer Institute, Louisville, KY 40202, USA; 2Division of Surgical Oncology, Department of Surgery, University of Louisville, Louisville, KY 40202, USA

**Keywords:** exocrine pancreatic insufficiency, pancreatectomy, malabsorption

## Abstract

Background/Objectives: This study explores the optimization of exocrine pancreatic insufficiency (EPI) management in pancreatic adenocarcinoma patients, focusing on the scientific advancements and technological interventions available to improve patient outcomes, including oral pancreatic enzyme replacement therapy (PERT) and immobilized lipase cartridge (RELiZORB^®^). This was a prospective Institutional Review Board (IRB)-approved study from October 2019 through to August 2021 at the Louisville Medical Center in collaboration with Norton Healthcare and the University of Louisville Division of Surgical Oncology. Patients with a diagnosis of pancreatic adenocarcinoma (Stage 2 or 3) who underwent oncologic surgical resection were included in this study. Methods: Patients were contacted at pre-defined intervals (prior to surgery, before hospital discharge, and 2, 4, 6, and 12 weeks after surgery) to complete nutrition evaluation, EPI assessment, and quality of life questionnaires to identify the severity and frequency of gastrointestinal (GI) symptoms. Results: EPI symptoms were reported in 28 of the 35 total patients studied (80%). Jejunostomy tubes were placed during oncologic surgery in 25 of the 35 total patients studied (71%), and 12 of the 25 patients with a jejunostomy tube utilized enzyme cartridges to manage EPI symptoms while on supplemental tube feeding (48%). EPI symptoms were reported in 8 of the 10 patients without a feeding tube (80%), and their EPI symptoms were managed with PERT alone. EPI interventions, both oral PERT and immobilized cartridges, were associated with a decrease in EPI symptoms after surgery and improved quality of life (QOL). Conclusions: Overall, early optimization of EPI is crucial to enhance overall patient care, return to oncology therapy after surgery, and improve quality of life in pancreatic adenocarcinoma patients.

## 1. Introduction

Pancreatic cancer is a highly morbid disease process that induces a number of gastrointestinal side effects and often interferes with the absorption of nutrients [1]. Patients with pancreatic cancer often experience symptoms related to malabsorption, such as bloating, gas, abdominal cramping, loose stools, frequent stools, and unintentional weight loss. These symptoms may present prior to diagnosis, during neoadjuvant chemotherapy treatment, or following surgical resection. Tumor location and size, ductal involvement, volume of chemotherapy received, and the degree of surgical resection all influence the degree of malabsorption [2]. When a tumor involves the pancreatic duct or disables the acinar cells, patients experience malabsorption related to exocrine pancreatic insufficiency (EPI). Tumors that involve the pancreatic head and body are more likely to obstruct the pancreatic duct and cause EPI symptoms prior to surgical resection. The prevalence of EPI following surgical resection is greater with pancreaticoduodenectomy (32%) than with distal pancreatectomy (11%) or central pancreatectomy (5%) [3].

The optimal completeness of oncologic care for resectable (Stage 1 and Stage 2) pancreatic cancers is at least 3 months (six cycles) of neoadjuvant/adjuvant chemotherapy in conjunction with definitive surgical margin negative pancreatectomy with adequate lymph node staging. Critical to the success of this care are prehabilitation efforts, which should be implemented to maintain physical performance/fitness and optimize and maintain nutritional status during the neoadjuvant treatment phase to increase tolerance to chemotherapy, avoid dose-limiting toxicity, and prepare patients for surgical resection/therapy. Efficient rehabilitation after pancreatectomy is critical to a patient’s oncologic care, as extended recovery can delay recommended adjuvant treatment. Optimal nutrition can be performed through the oral route, the enteral route (gastrostomy tube, jejunostomy tube, or naso-jejunal tube), or a combination of both. Patients with EPI symptoms are particularly challenging to manage when a portion or entirety of their nutrition is provided through jejunostomy feedings [3]. This study provides data to support clinical decision making and guidance for pancreatectomy patients with EPI symptoms. We also provide a framework for subjectively diagnosing EPI and guidance for how to effectively utilize a combination of pancreatic enzyme replacement strategies, including immobilized lipase cartridges in connection with enteral pump feedings.

## 2. Materials and Methods

This was a prospective Institutional Review Board (IRB)-approved study that was conducted from October 2019 through to August 2021 at the Louisville Medical Center in collaboration with Norton Healthcare and the University of Louisville Division of Surgical Oncology. Prospective patients who were undergoing pancreatectomy (Stage 2 and Stage 3) and/or irreversible electroporation (IRE) (Stage 2B or Stage 3) gave consent and were enrolled in this treatment algorithm [4,5]. Patient selection, staging, neoadjuvant treatment, and the surgical technique have all been previously reported [4,6]. Neoadjuvant chemotherapy or radiation therapy were not required for study inclusion. In short, staging included a triple-phase computed tomographic (CT) scan with less than 1.5 mm cuts at the time of diagnosis and repeated 1–2 weeks prior to resection. After induction therapy, patients underwent pancreatectomy alone, pancreatectomy with IRE, or IRE alone in the operating room using open or laparoscopic techniques previously reported. In patients who received neoadjuvant chemotherapy, FOLFIRINOX, gemcitabine and abraxane, or single-agent gemcitabine were the only induction regimens acceptable for inclusion (Table 1). Proton pump inhibitors were only prescribed preoperatively in response to symptoms of gastroesophageal reflux disease and in all patients postoperatively to help prevent upper GI bleeding and stress ulcers.

Patients were screened in the outpatient clinic space after presenting with a diagnosis of a pancreatic mass with or without positive biopsy for malignancy. Eligible patients were included in this study after (1) receiving a biopsy-proven diagnosis of pancreatic malignancy and (2) deemed eligible for surgical resection or surgical intervention related to pancreatic cancer diagnosis. Unresectable pancreatic cancer patients were excluded from this study. Patients with a benign pancreatic process were excluded from this study. All patients were evaluated for EPI symptoms before and after surgical intervention.

Preoperative nutrition status is an important indicator for postoperative nutrition performance. Patients with significant weight loss (>10%) during neoadjuvant therapy, a preoperative prealbumin level <14 mg/dL, a pulmonary comorbidity (defined as pulmonary function testing <70% predicted and/or preoperative supplemental oxygen requirement), or significant vascular reconstruction during surgery were considered for a jejunostomy tube (J-tube) placement at the time of surgical resection [7]. Early identification and management of digestive symptoms, such as EPI, helps to optimize preoperative nutrition performance and enhanced postoperative recovery.

### 2.1. EPI Evaluation and Diagnosis

Fecal elastase testing has been widely accepted as the gold standard for clinical EPI diagnosis; however, reliance on this test alone presents a challenge to patient care [3,4]. Fecal elastase testing is unreliable in the setting of diarrhea because watery stools can dilute the elastase and give a falsely low result. Fecal elastase testing is not always a covered laboratory service, resulting in an out-of-pocket expense to the patient. More importantly, not all laboratories are capable of processing this test, requiring external testing which delays time to receive a result. While fecal elastase testing remains the preferred diagnostic marker for identifying EPI, it is not required to obtain a diagnosis. EPI can successfully be diagnosed based on clinical symptoms alone [3].

For this study, EPI was identified subjectively without definitive testing using any of the established diagnostic studies [8]. Our laboratory does not process fecal elastase testing internally; all tests are sent for outside review with an average 7–10-day turnaround time to obtain a result. Our experienced team was able to successfully diagnose EPI based on relevant symptoms alone. The criteria utilized were either subjective symptoms of malabsorption including steatorrhea greater than three episodes per day occurring at least three days per week or unexplained weight loss of more than 10% of preoperative weight that improved with PERT. Other symptoms that were used for confirmation were epigastric or colicky abdominal pain, bloating, and frequent stools. Not all patients with EPI will experience steatorrhea, which is present only when >90% acinar cell function is lost due to the inadequate delivery of both lipase and bicarbonate. The consideration of other relevant EPI symptoms is recommended to avoid delaying diagnosis and intervention.

EPI symptoms were most common beginning one to four hours after connecting to enteral feedings and were ongoing with continuous feeding regardless of formula or rate adjustments. The question was raised as to whether or not the addition of oral PERT would address new symptoms of fat malabsorption from enteral nutrition. Given the contraindication to open, crush, and bolus PERT capsules through a J-tube, patients with the ability to safely take oral medications were asked to take one Creon^®^ 36,000-unit capsule by mouth four times in 24 h for possible EPI management. EPI patients with oral diets were also instructed to take one Creon^®^ 36,000-unit capsule prior to all oral meals and snacks. Outcomes were recorded at interval follow-up discussions utilizing two gastrointestinal quality of life questionnaires and follow-up interviews with an oncology registered dietitian [9,10].

Two algorithms were developed and implemented in order to standardize EPI management and interventions with and without oral diet orders while on enteral feedings [3]. Enteral nutrition patients with oral diet orders were given PERT for 24 h and then re-evaluated for ongoing or resolved EPI symptoms. If symptoms persisted after 24 h of EPI management with oral PERT alone, immobilized lipase cartridges were added to provide EPI management for tube feeding.

### 2.2. Immobilized Lipase Cartridges

Commercially available enteral formulas in the United States do not contain hydrolyzed fat secondary to instability with extended storage times; therefore, triglycerides in enteral nutrition must be hydrolyzed at the time of feeding when EPI is present. For the purpose of this study, we chose to use immobilized lipase cartridges made by Alcresta Therapeutics, trade name RELiZORB^®^, based out of Waltham, MA, USA, to hydrolyze triglycerides in enteral formula at the time of feeding. Fat hydrolysis occurs as the formula flows through the cartridge, delivering free fatty acids into the digestive tract that are easily absorbed with or without the presence of pancreatic enzymes. One cartridge contains enough lipase to hydrolyze the fat in up to 500 mL of enteral formula. Two cartridges can be connected in tandem to provide enough hydrolysis for up to 1000 mL of enteral formula. Patients were given samples of study cartridges to use while enrolled in the study, and compliance was monitored at each follow-up interval. RELiZORB^®^ is the only device currently available to hydrolyze fat in enteral formula at the time of feeding. Alternative regimens include fat-free formula substitutions or a transition to parenteral nutrition.

### 2.3. Quality of Life Questionnaire

GI Tolerance Short and Gastrointestinal Quality of Life Index (GIQLI) questionnaires were used to identify the severity and frequency of symptoms, and the degree to which symptoms interfere with daily quality of life [9,10]. The registered dietitian contacted study patients prior to surgery, prior to hospital discharge, and two, four, six, and twelve weeks following surgery to complete routine assessment and education, as well as complete both quality of life questionnaires. All assessments were performed in person or by telephone with the registered dietitian. We observed 89.4% compliance with QOL questionnaire completion.

## 3. Results

A total of 35 patients completed preoperative QOL questionnaires and dietitian assessment. In addition, patients with preoperative EPI diagnosis received education regarding their current EPI management, as well as a review of postoperative expectations. Enteral nutrition patients with a preoperative EPI diagnosis were managed postoperatively using the algorithm in Figure 1. Enteral nutrition patients with no preoperative EPI diagnosis were managed postoperatively using the algorithm in Figure 2.

Symptoms were reported in 28 of the 35 total patients studied (80%): only 1 patient reported preoperative EPI symptoms alone (2.9%), 11 patients reported postoperative EPI symptoms alone (31%), and 16 patients reported EPI symptoms both before and after surgery (45%). J-tubes were placed in 25 of the 35 total patients studied (71%). In the J-tube group, 20 patients reported EPI symptoms (80%), 12 patients utilized enzyme cartridges to manage EPI symptoms (48%), 4 patients managed EPI symptoms with PERT alone (16%), 4 patients stopped tube feeding after exhibiting EPI symptoms (16%), and 5 patients did not exhibit EPI symptoms (20%). The four patients who managed EPI symptoms with PERT alone also stopped using PERT before the 12-week follow-up, which suggests their GI symptoms were likely unrelated to EPI. EPI symptoms were reported in 8 of the 10 patients without a J-tube (80%), and their EPI symptoms were managed with PERT alone.

The results were analyzed based on symptom presence before and after surgery. Standard deviation and a *t*-test were used for statistical analysis. Reported instances of diarrhea, urgency, overnight stools, early morning stools, frequent stools, and gurgling decreased after initiation of enzyme cartridges (Figure 3 and Table 2).

One patient who reported an increase in diarrhea, frequent stools, and overnight stools was diagnosed with C. difficile infection. Reported instances of excess gas, bloating, fullness, nausea, pain, and uncontrolled stools slightly increased after initial initiation of enzyme cartridges (Figure 4).

Decreased ambulation following surgery has been associated with increased gas, bloating, and fullness due to decreased transit time and GI motility. Patients with persistent symptoms of fullness, nausea, and bloating were encouraged to increase the duration and frequency of ambulation for improved GI motility and digestion. Patients with persistent symptoms of gas, bloating, and uncontrolled stools were prescribed prebiotic fiber and high-potency probiotics to encourage gut microbiome restoration following neoadjuvant chemotherapy and surgical intervention. No enzyme cartridges were used in the preoperative setting.

### 3.1. Compliance with Lipase Cartridges

While we made every effort to verify compliance with enzyme cartridge use at all interaction points, quantifying compliance was difficult. Most patients were only able to estimate the number of remaining cartridges at assessment intervals and some did not monitor supply usage at all. One patient in the study stopped enteral nutrition without notifying providers. One patient in the study was re-admitted to the hospital at 6 weeks postoperatively and diagnosed with COVID-19 infection, which ultimately led to his demise. Most patients reported tandem use of enzyme cartridges (2 per 1000 mL formula infused) and admitted to changing cartridges with each new 1000 mL of formula delivered. No patients reported difficulty with enzyme cartridge setup or use.

### 3.2. Multifactoral Symptom Management

Symptoms of oncologic treatment often mask symptoms of fat malabsorption, resulting in an underdiagnosis of EPI in oncology patients [3]. Diagnostic testing for EPI, such as fecal elastase testing, can be particularly challenging for frail oncology patients with liquid or watery stools or while actively receiving systemic treatment. The difficulties with obtaining a proper stool sample in symptomatic oncology patients contributes to the poor compliance with EPI evaluation and diagnosis. Additionally, some facilities send out all fecal elastase tests, which further delays the response time. While subjective testing based on symptoms alone carries potential bias and an increased risk for underestimation or overestimation of EPI, subjective evaluation is often the most appropriate method for the oncology population.

Pancreatic cancer patients are often heavily pre-treated with neoadjuvant chemotherapy and in some cases radiation therapy, which increases the risk of malabsorption from microbiome abnormalities, bacterial overgrowth, and GI mucositis. Symptoms from oncologic treatment often induce side effects similar to EPI, making it difficult for practitioners to distinguish between EPI symptoms and treatment toxicity. In cases where we observed symptom improvement but not complete resolution after initiation of lipase cartridges, patients were concurrently treated with refrigerated high-potency probiotics, prebiotic fiber, and in some cases, anti-diarrheal medication. It is certainly plausible that pancreatic cancer patients who have undergone a neoadjuvant systemic course followed by surgical intervention will have some degree of malabsorption unrelated to EPI. Therefore, it is reasonable and recommended to manage postoperative malabsorption with multiple interventions and therapies (Figure 1 and Figure 2).

Initiation of lipase cartridges for EPI management in patients with supplemental enteral nutrition demonstrated clear benefits, which is why we now routinely use this device in practice. PERT alone did not appear to effectively treat EPI symptoms in patients who were receiving sole-source or supplemental enteral nutrition. Therefore, we do not feel that it is appropriate to prescribe PERT alone for EPI management in patients who are receiving enteral nutrition. Concurrent use of PERT and enzyme cartridges is recommended for patients receiving supplemental enteral nutrition, and education regarding simultaneous use of PERT and enzyme cartridges is necessary to ensure patient compliance.

## 4. Discussion

The management of EPI and intestinal malabsorption in pancreatic cancer patients includes a range of treatment options, such as changes in diet composition, changes in eating habits to avoid dumping symptoms, pancreatic enzyme replacement therapies, anti-diarrheal medications such as Imodium (Loperamide HCl) or Lomotil (Diphenoxylate/Atropine sulfate), prebiotic fiber, and high-potency probiotics. The key foundational strategy for EPI management is pancreatic enzyme replacement therapy (PERT) which aims to replenish the vital digestive enzymes necessary for optimal nutrient absorption [11].

The different oral pancreatic enzyme preparations available include Creon^®^, ZenPep^®^, Nutrizym^®^, Pancreaze^®^, and Pancrex^®^. These pancrelipase medications are porcine-derived and commonly come in varying capsule sizes, ranging from 3000 to 40,000 units per dose. We chose to use Creon^®^ 36,000-unit capsules for the patients in this study. PERT capsules must be taken moments before starting a meal or snack to be effective and repeated if meals last longer than 60 min [3]. PERT loses efficacy if taken after the start of a meal or at the end of a meal [12]. Simultaneously with PERT, nutritional support and diet education are critical to ensuring an optimal recovery after pancreatic resection and/or electroporation [13].

Additionally, the integration of novel drug delivery systems has become instrumental in refining the administration and effectiveness of therapeutic interventions [12]. Embracing the era of personalized medicine, tailored approaches are being explored to customize treatments based on individual patient profiles, ensuring a more precise and effective response to EPI. Beyond these core oral interventions, the use of new pancreatic enzyme replacement therapy with enteral feeds has also grown in recent years.

This study demonstrated that oral enzyme replacement therapy alone did not properly address fat malabsorption in patients receiving enteral tube feeding. Due to the brief intervention time of oral PERT (effective only up to 60 min after each use) and the disparagement of crushing and administering the medication through a feeding tube, oral enzyme replacement therapy should not be implemented for the purpose of fat hydrolysis of enteral formula. This study demonstrated that lipase cartridges concurrent with enteral pump feeding was the only effective management of EPI in patients with J-tube feeding.

Pancreatic cancer patients treated with neoadjuvant chemotherapy and/or radiation therapy were more likely to demonstrate nutrient malabsorption after surgery [14]. While the addition of pancreatic enzyme replacement therapies offered some relief of symptoms, these patients also benefitted from interventions intended to improve gut integrity and regenerate a healthy gut microbiome [15]. Additional interventions such as prebiotic fiber, high-potency probiotics, and in some cases, intravenous hydration and anti-diarrheal medication therapy should be implemented in combination with enzyme replacement therapy to improve nutrient absorption in these fragile patients. Close follow-up with a specialized oncology dietitian is recommended to ensure these patients maintain their weight and nutrition status when adjuvant therapy is recommended [16].

Two patients in this study switched from supplemental enteral nutrition therapy to total parenteral nutrition therapy <30 days after surgery following multiple failed attempts to address postoperative malabsorption. These patients also demonstrated early recurrence (<90 days postoperative) of their pancreatic cancer, which we suspect was the cause of ongoing malabsorption and GI symptoms. It has been our experience that nutrient absorption even in fragile patients improves with a combination of therapeutic interventions (enzyme replacement, dietary, and microbiome regenerating) and the rare need for total parenteral nutrition. This integrated approach underscores the necessity of addressing the multifaceted challenges associated with EPI in the specific context of pancreatic cancer, fostering a more fluid and adaptable framework for patient care.

In the continuous refinement of EPI management, emerging strategies are contributing to more effective approaches. Targeted therapies represent a significant advancement, tailoring interventions to specific aspects of EPI pathology for enhanced precision and efficacy [17]. Additionally, recognizing the role of microbiota modulation has become integral, as understanding and influencing the gut microbiome can impact pancreatic function [15]. The identification and utilization of biomarkers for early detection and continuous monitoring of EPI are crucial elements in proactive management [18]. Moreover, addressing EPI-related complications is a key focus, emphasizing a comprehensive approach to mitigating potential health challenges associated with this condition [19]. These emerging strategies collectively underscore the dynamic and evolving landscape in the optimization of EPI management.

While advancements are being made in optimizing EPI management, several challenges and limitations persist. One significant obstacle is the issue of adherence to treatment protocols, as ensuring consistent and proper implementation of therapeutic measures remains a considerable challenge. Economic considerations pose another hurdle, with the cost implications of long-term management strategies potentially limiting accessibility for some individuals [20]. Additionally, the absence of comprehensive and standardized clinical guidelines creates a further challenge, as healthcare providers may face difficulties in navigating the complex landscape of EPI management without clear directives [21]. Addressing these challenges is pivotal for the successful optimization of EPI management, necessitating a concerted effort to enhance adherence, consider economic factors, and develop robust clinical guidelines for more effective and equitable patient care.

The authors acknowledge several limitations of this study, including the small sample size, subjective diagnosis of EPI, and limited number of publications reviewing combined surgical procedures with both pancreatectomy and irreversible electroporation (IRE). First, the date range for this study overlapped the onset and peaks of the COVID-19 pandemic, which complicated patient enrollment. Our institution observed an increase in the number of patients presenting with advanced-stage cancers, presumably due to the avoidance of healthcare facilities during the pandemic [22]. Second, subjective diagnosis of EPI introduces diagnostic bias and variability; however, the management of fat malabsorption based on subjective findings and provider clinical judgment has proven to be an effective strategy when diagnostic testing is unavailable or untimely [23]. Finally, pancreatectomy plus surgical margin IRE is a relatively novel surgical intervention with limited published data. This combined procedure has been performed at our institution for over a decade with observed similarities in postoperative symptoms between IRE, pancreatectomy, and pancreatectomy-plus-IRE patients. Thus, we felt it was appropriate to include all three procedure types in this study.

## 5. Conclusions

In conclusion, the management of EPI is a multifaceted endeavor marked by both progress and persistent challenges. The summary of the key findings underscores the importance of technological and emerging strategies, such as improved formulations of PERT, targeted therapies, microbiota modulation, and biomarkers for early detection. However, challenges such as treatment adherence, economic considerations, and the lack of comprehensive clinical guidelines must be addressed to optimize EPI management fully. Looking ahead, future directions should involve continued research to refine existing strategies, explore novel interventions, and develop clearer clinical guidelines. Recommendations include fostering patient education to enhance adherence, addressing economic barriers, and promoting collaborative efforts between healthcare providers and researchers. By acknowledging these findings and embracing future directions, there is potential for significant advancements in the comprehensive and effective management of EPI.

## Figures and Tables

**Figure 1 nutrients-16-03499-f001:**
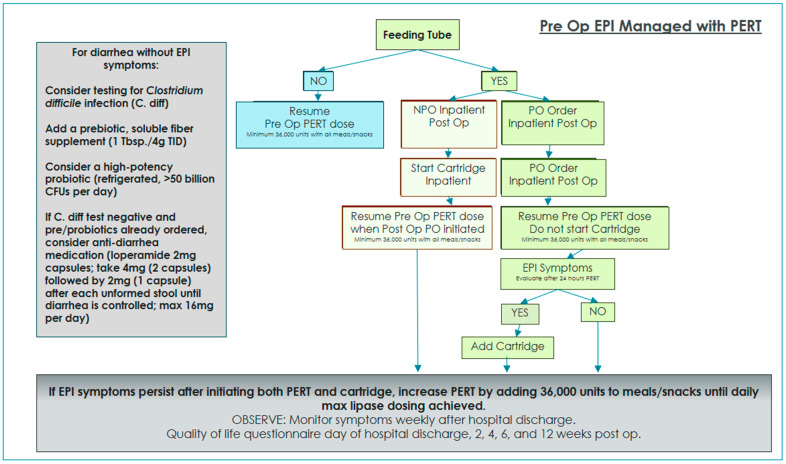
The EPI recommended treatment algorithm for a pancreatic cancer patient with or without a gastrostomy or a jejunal feeding tube with documented preoperative EPI symptoms. Abbreviation: colony-forming units (CFUs).

**Figure 2 nutrients-16-03499-f002:**
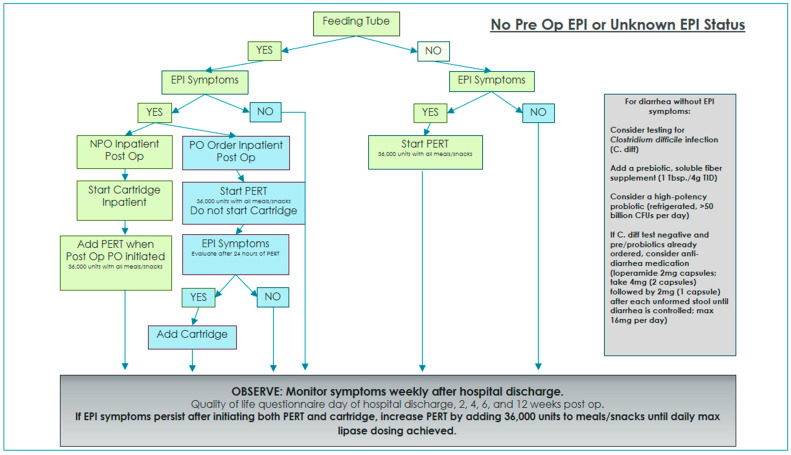
The EPI recommended treatment algorithm for a pancreatic cancer patient with or without a gastrostomy or a jejunal feeding tube with no preoperative EPI symptoms. Abbreviation: colony-forming units (CFUs).

**Figure 3 nutrients-16-03499-f003:**
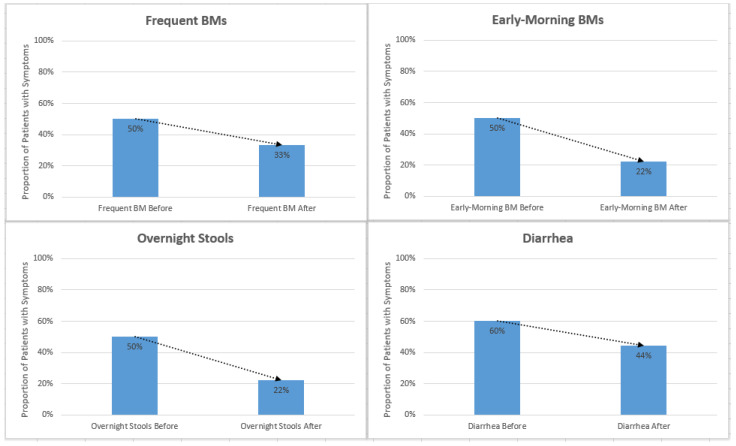
Reported instances of diarrhea, urgency, overnight stools, early-morning stools, frequent stools, and gurgling decreased after initiation of enzyme cartridges. Abbreviation: bowel movement (BM).

**Figure 4 nutrients-16-03499-f004:**
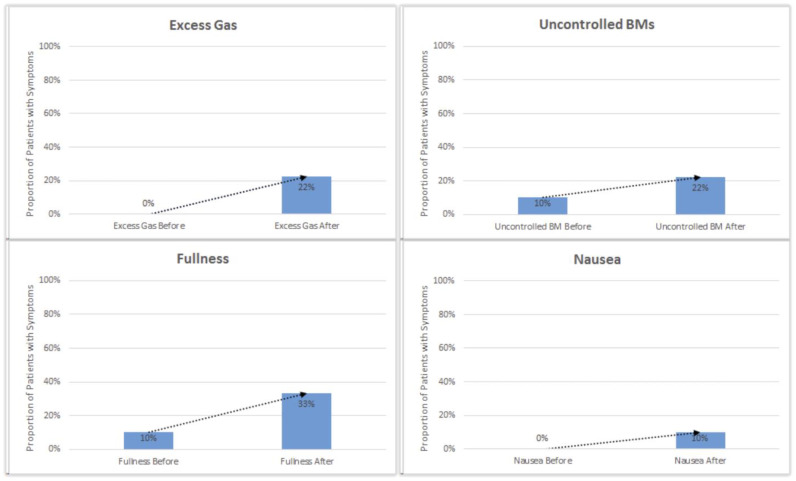
Reported instances of excess gas, bloating, fullness, nausea, pain, and uncontrolled stools slightly increased after initial initiation of enzyme cartridges. Abbreviation: bowel movement (BM).

**Table 1 nutrients-16-03499-t001:** Patient demographics.

	Tube-Fed Patients (n = 25)	Non-Tube-Fed Patients (n = 10)
Demographics		
Age in years (mean)	65	63.5
Gender		
Male	15	4
Female	10	6
Pancreatic Tumor Location		
Head	15	7
Neck/Mid-Body	7	0
Tail	2	2
Ampulla	0	1
Duodenal/Bile Duct	1	0
Preoperative Chemotherapy		
Yes	21	9
No	4	1
Median (Range)	6 months (2–14)	5.5 months (3–8)
Preoperative Radiation Therapy		
Yes	4	4
No	21	6
Surgical Approach		
Open	4	1
Laparoscopic	21	9
Surgical Procedure		
Pancreaticoduodenectomy	8	7
Distal Pancreatectomy	7	0
Distal Pancreatectomy + IRE	1	0
Hepaticojejunostomy + Gastrojejunostomy + IRE	3	1
Irreversible Electroporation Alone	6	1
Completion Pancreatectomy	0	1

**Table 2 nutrients-16-03499-t002:** Patient-reported symptoms.

	Patient-Reported Symptoms Before Starting Relizorb within One Week of Pancreatic Surgery		Patient-Reported Symptoms after Starting Relizorb within One Week of Starting Relizorb		*p*-Value
Symptom	Occurrence	Standard Deviation	Occurrence	Standard Deviation	
Overnight Stools	5	0.53	2	0.44	0.23
Early-Morning Stools	5	0.53	2	0.44	0.23
Abdominal Pain	2	0.42	2	0.44	0.91
Fullness	1	0.32	3	0.50	0.24
Bloating	1	0.32	1	0.33	0.94
Excess Gas	0	0	2	0.44	0.13
Gurgling	4	0.52	2	0.44	0.43
Frequent Stools	5	0.53	3	0.50	0.49
Urgent Stools	7	0.48	4	0.53	0.29
Diarrhea	6	0.52	4	0.53	0.52
Nausea	0	0	1	0.32	0.36
Uncontrolled Stools	1	0.32	2	0.44	0.49

## Data Availability

The raw data supporting the conclusions of this article will be made available by the corresponding authors on request.

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
