# Peer review of "Optimization of Exocrine Pancreatic Insufficiency in Pancreatic Adenocarcinoma Patients"

_nutrients, 2024, doi:10.3390/nu16203499_

Round 1
Reviewer 1 Report
Comments and Suggestions for Authors
The Abstract should be more focused making clear that all patients were evaluated preoperatively for pancreatic exocrine insufficiency and jejunostomy feeding tubes were placed during surgery. In addition the time points of postoperative EPI evaluation and the effectiveness of replacement therapies should be mentioned.
Materials and Methods
Please clarify that all patients received neo-adjuvant chemotherapy and/or radiation therapy before surgery (line 71) as there is some confusion with the data shown in Table 1.
Did the authors prescribe routinely proton pump inhibitors to all patients for preventing inactivation of Creon by the gastric acidity?
Discussion
The discussion generally presents thoughts and opinions of the authors and there is little use of relevant literature.
Other minor points
Please make sure to give abbreviations in full when first mentioned. See: EPI. IRB, GI, J tubes and QOL (Abstract), IRB (line 63), PFT (line 84), PO (line 122, 124, 129, 130 probably oral can be used as an alternative) and GI Tolerance Short and GIQLI gastrointestinal QOL (line 150), Hepaticoj + Gastroj (Table 1).
Please add years for age in Table 1
Explanatory comments for the abbreviations used in Figures 1 and 2 might be useful. The same applies for BM in Figures 3 and 4.
Author Response
Note: All edits to the manuscript are in red text.
Comment 1:
The Abstract should be more focused making clear that all patients were evaluated preoperatively for pancreatic exocrine insufficiency and jejunostomy feeding tubes were placed during surgery. In addition the time points of postoperative EPI evaluation and the effectiveness of replacement therapies should be mentioned.
Response 1:
Thank you for pointing this out. We have accounted for all suggestions and updated lines 11, 12, 19, 20, 22, 23, 27, and 28 based on your feedback.
Comment 2:
Materials and Methods
Please clarify that all patients received neo-adjuvant chemotherapy and/or radiation therapy before surgery (line 71) as there is some confusion with the data shown in Table 1.
Did the authors prescribe routinely proton pump inhibitors to all patients for preventing inactivation of Creon by the gastric acidity?
Response 2:
Thank you for pointing this out. Lines 71, 72, and 76 have been updated based on your feedback.
Lines 78-80 have been updated to include description of PPI utilization.
Comment 3:
Discussion
The discussion generally presents thoughts and opinions of the authors and there is little use of relevant literature.
Response 3:
Thank you for pointing this out. Additional supporting references have been added to lines 271 and 346.
Comment 4:
Other minor points
Please make sure to give abbreviations in full when first mentioned. See: EPI. IRB, GI, J tubes and QOL (Abstract), IRB (line 63), PFT (line 84), PO (line 122, 124, 129, 130 probably oral can be used as an alternative) and GI Tolerance Short and GIQLI gastrointestinal QOL (line 150), Hepaticoj + Gastroj (Table 1).
Please add years for age in Table 1
Explanatory comments for the abbreviations used in Figures 1 and 2 might be useful. The same applies for BM in Figures 3 and 4.
Response 4:
Thank you for pointing this out. Abbreviations for EPI, IRB, GI, J tubes, and QOL have been updated throughout the abstract (lines 12, 15, 21, 22, 24, 28). Abbreviations were updated throughout the body: IRB (line 65), PFTs (lines 91-92), all PO abbreviations were substituted with “oral” (lines 129, 131, 136, 137), GIQLI was expanded (line 157), and Table 1 was updated (Hepaticojejunostomy + Gastrojejunostomy and Age in years).
Explanatory comment for CFU was added to Figure 1 and 2. All other abbreviations are previously noted in the text. Explanatory comment for BM was added for Figure 3 and 4.
Reviewer 2 Report
Comments and Suggestions for Authors
Relying on symptoms alone for EPI diagnosis introduces potential bias and variability. Incorporating objective measures like fecal elastase testing would strengthen the diagnostic criteria.
The study doesn't clearly describe how patients were selected, which could lead to bias in the results.
The study acknowledges difficulties in quantifying compliance with enzyme cartridge use. Implementing a more rigorous method to track compliance would improve data reliability.
The study doesn't mention any blinding procedures, which could introduce bias in symptom reporting and assessment.
The article doesn't provide detailed information about the statistical analyses used, making it difficult to assess the validity of the results.
Author Response
All edits to the manuscript are in red text.
Comment 1:
Relying on symptoms alone for EPI diagnosis introduces potential bias and variability. Incorporating objective measures like fecal elastase testing would strengthen the diagnostic criteria.
Response 1:
Thank you for pointing this out. All authors are aware that diagnosis based on symptoms alone allows for potential bias. Our rational for proceeding without fecal elastase testing is described in the manuscript (lines 103-123 and lines 347-350).
Comment 2:
The study doesn't clearly describe how patients were selected, which could lead to bias in the results.
Response 2:
Thank you for pointing this out. Another reviewer shared a similar concern which prompted edits to the methods section (lines 71, 72, 76, and 77).
Comment 3:
The study acknowledges difficulties in quantifying compliance with enzyme cartridge use. Implementing a more rigorous method to track compliance would improve data reliability.
Response 3:
Thank you for pointing this out. All authors completely agree with this suggestion and will take into account for future studies.
Comment 4:
The study doesn't mention any blinding procedures, which could introduce bias in symptom reporting and assessment.
Response 4:
Thank you for pointing this out. All authors acknowledge the risk for bias in symptom reporting and assessment without blinding; however a blinded design was not available. All authors appreciate the suggestion and will take into account for future studies.
Comment 5:
The article doesn't provide detailed information about the statistical analyses used, making it difficult to assess the validity of the results.
Response 5:
Thank you for pointing this out. An update to address this concern has been to the results section (lines 197, 198).